# Evaluating the Use of QR Codes on Food Products

**Konstantinos Rotsios** [1], **Aggeliki Konstantoglou** [2], **Dimitris Folinas** [3,*], **Thomas Fotiadis** [2], **Leonidas Hatzithomas** [4] and **Christina Boutsouki** [5]

1. Department of International Business, Perrotis College, American Farm School, 551 02 Thessaloniki, Greece; krotsi@afs.edu.gr
2. Department of Production Engineering and Management, Democritus University of Thrace, 671 00 Xanthi, Greece; angiekonsta@gmail.com (A.K.); dr.fotiadis.thomas@gmail.com (T.F.)
3. Department of Supply Chain Management, International Hellenic University, 601 00 Katerini, Greece
4. Department of Business Administration, University of Macedonia, 546 36 Thessaloniki, Greece; hatzithomas@uom.edu.gr
5. Faculty of Economic and Political Sciences, Aristotle University of Thessaloniki, 541 24 Thessaloniki, Greece; chbouts@econ.auth.gr
* Correspondence: folinasd@ihu.gr; Tel.: +30-69-3831-2524

**Abstract:** Today, consumers consider food packaging to be as equally important as a product brand. In addition, the increase in smartphone usage by consumers has led marketers to design new forms of packaging. Among the latest marketing trends, smart packaging with the use of QR Codes is emerging as one of the most promising technologies to enhance the information provided to consumers and influence their buying behavior. This study evaluates the use of a QR Code on bottled milk and more specifically on milk produced by one of the most well-known "boutique" Greek dairy producers. It consists of two phases. In the first one, data was gathered from 537 consumers of the product to capture and analyze their (i) buying behavior, (ii) perception of the product's package, and (iii) knowledge about the product. In the second phase, a Quick Response (QR) Code was placed on the bottle's label. Consumers who scanned it were linked to a web page containing information on the product. A total of 308 from the 537 initial respondents scanned the code, accessed the site, and answered the second questionnaire. Similar to the first stage, (i) the consumers' buying behavior, (ii) their perception of the product's package, and (iii) their knowledge about the product were examined, following their visit to the above-mentioned website through the QR Code. The objective was to evaluate the use of web applications using enriched text information. The results show that a QR Code on the packaging of food products, which directs consumers to entertaining and enriched content, results in an increased level of usage intention. Moreover, they proved that comprehension and self-confidence are higher with the adoption of the QR Code. In addition, the use of QR Codes enables businesses to provide timely and accurate information and positively influence consumers' buying behavior.

**Keywords:** packaging; QR code; food products; consumer behavior

## 1. Introduction

Product packaging has become increasingly important as it stimulates impulsive buying behavior, increases market share, and reduces promotional costs [1]. One of the most comprehensive definitions of packaging was proposed by Saghir, who defined packaging as a "*coordinated system of preparation of goods for safe, efficient and effective handling, transport, distribution, storage, retail, consumption and recovery, reuse or marketing combined with maximizing consumer value, sales and therefore profit*" [2].

In addition to the functional role, packaging plays a communicative role in terms of image, product, and producer identity [3]. According to Shimp, packaging can be described as an affordable form of promotion, an effective salesperson, and a brief advertisement [4].

Furthermore, Silayoi and Speece argued that packaging becomes a critical factor of the in-store consumer decision-making process [5,6].

Packaging is one of the most crucial components of the food industry. Almost all food products traded and consumed are available in some form of packaging. An important part of the packaging is the label that most packages carry. The packaging label is defined by Robertson as *"any written, electronic or graphic form of communication on the package or a separate but integrated sticker on the product"* [7].

The size of the global packaged food market was estimated at $303.26 billion in 2019, with a compound annual growth rate (CAGR) of 5.2% over this period [8]. Research results on the "power" of packaging as a means of promoting and influencing consumer purchasing behavior are notable. A study by the Paper and Packaging Board and IPSOS shows that 7 in 10 (72%) consumers in the USA agree that packaging design can impact their purchase [9]. Additionally, based on the same research, consumers consider packaging to be as important as the product brand (in percentages of 10–12%, respectively). Surveys have also shown that companies that pay close attention to their packaging often report a 30% increase in consumer interest [10].

The rapid growth of information and communication technologies and the knowledge-based economic evolution are driving organizations to utilize technological approaches to business activities. The increase in smartphone use by consumers has led marketers to design new forms of packaging. Among the latest marketing trends, smart packaging with the incorporation of QR Codes is emerging as one of the most promising technologies to increase the information provided to consumers and influence their buying behavior. According to Albastroiu and Felea, QR Codes allow quick access to information such as website addresses, e-mails, phone numbers, geographical coordinates, etc., through mobile devices [11]. They can be used on product labels and advertising media and consumers can access their content with any mobile or smartphone with a built-in camera and QR Code reader software [12].

Previous research has examined the use of QR Codes on the packaging and its impact on online marketers. For example, studies have examined consumers' awareness and acceptance of QR Codes [13,14], while others have investigated the potential application of QR Codes in various business activities, such as trade [15], marketing [16], and supply chain [17]. Furthermore, a large number of surveys have been conducted, recording consumers' perception regarding the usage of QR Codes in food products. According to the results of a survey [18] titled QR Code statistics 2022: Latest numbers and use-cases on global usage:

(1)  57% scanned a food QR Code to get specific information about the product.
(2)  38.99% of respondents want to see QR Codes used more broadly in the future.
(3)  67% of the respondents agreed that these codes make life easier.

In this, we study aimed to examine the effectiveness of the use of QR Codes on consumers' behavior for a specific product, the bottled milk produced by the American Farm School of Thessaloniki, Greece (www.afs.edu.gr/products/, accessed on 1 December 2021), and on a specific market segment, Northern Greece. By introducing a new label supported by a QR Code, as well as by creating and launching a dedicated website for the study (https://sxolistogala.gr/, accessed on 1 December 2021), the authors examined the degree to which QR Codes on bottled milk packaging enhance the information provided to consumers about the product and increase their purchasing intention.

This study proposes an extended TAM model by adding three more variables: perceived knowledge of the product, perceived communication or interactions with the company's website, and perceived feeling of security. This constitutes the base for the conceptual framework that explores consumers' intention to use QR Codes on packaging. Therefore, we aimed to evaluate the findings of the above studies using a real-life example case. The authors strongly believe that the findings of the primary research will help managers of the food industry consider including QR Code technology and e-commerce approaches in the packaging schemes of their products.

The remainder of the paper is organized as follows. First, the importance of packaging and labeling of food products is presented. Next, the importance of QR Codes on food labeling is discussed and the research questions are presented. In the following sections, the methodology is discussed and the results are presented. The paper closes with a discussion of the findings followed by the conclusions.

## 2. What Does Packaging Mean for Marketing?

Packaging and labeling of food products significantly affect consumer behavior in several ways, such as product display and shelf placement, image creation, product differentiation, and brand promotion. There are five main elements of the relationship between packaging and marketing [19–21] (Figure 1):

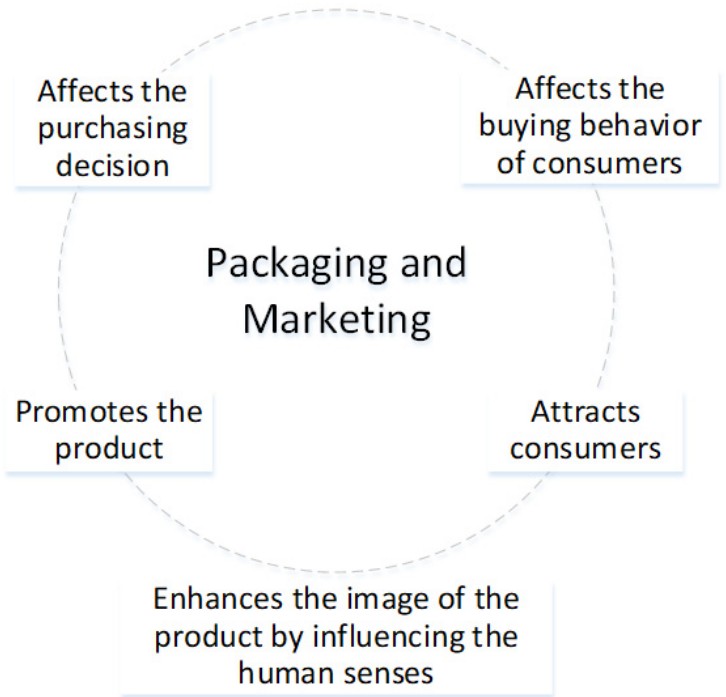

**Figure 1.** Correlation (relationship) between packaging and marketing.

1. Enhances the image of the product by influencing the human senses. Traditionally, marketing uses the human senses in several ways, seeking to provoke positive emotions and attitudes towards products. In regards to visual presentation, packaging strengthens the consumers' perception of the product and reminds consumers of it. This practically means that packaging provides images, shapes, logos, information, etc., that relate to a product and are received by consumers through sight. In addition, the sense of touch influences the consumers' perception of the food product. In terms of smell, it is the latest trend in creative multi-sensing marketing. Some packages use perfumes aiming at consumers' the sense of smell. Certain scents can evoke emotions, memories, and influence food-purchasing behavior [22].
2. Attracts consumers. A pleasant design, in addition to good quality materials, will attract the consumers' attention to examine the product closely, a critical step before the purchasing decision. The packaging of the food product on the shelf should differentiate the product and increase its appeal to consumers. Packaging should enable the consumer to: distinguish the product among other competitors, recognize the type of product, attract interest in the product, easily access instructions for use and information about the contents of the package, understand the usefulness of the product, and convince consumers to purchase it [23].

3.  Promotes the product. An eye-catching packaging attracts consumers, which is in accordance with the narrative promoted by the brand [24]. Essentially, it allows food producers to tell, in a very short time, their brand story and what the product stands for [25]. It is therefore reasonable to argue that packaging can also act as an in-store salesperson for businesses [5,6]. Packaging is considered a critical marketing tool because it creates a connection between the consumer and the product [26]. The image of the company and the brand name are enhanced through widely recognized packaging. Innovative packaging with clear benefits can result in significant profits. Several successful companies have managed to create such recognizable packaging that it has become part of their brand and communication strategy [27].

4.  Affects the buying behavior of consumers. Due to its multiple roles, packaging as part of the product itself includes almost all parts of businesses [28]. Meyers and Gerstman noted that consumers buy products based on what they see and read on the packaging [29]. Furthermore, it is common for customers to evaluate the product based on the packaging [27,30]. According to Orth, even less important packaging in terms of appearance, such as egg boxes, is evolving to deliver brand messages, differentiate from the competition, and attract consumers [31]. Additionally, Murad argued that packaging protects product content, promotes it to consumers, and provides useful information about the products contained, i.e., ingredients, storage temperature, date of production, etc. [32].

5.  Affects the purchasing decision. The role of packaging as part of the product is becoming increasingly important. Especially considering that 70% of purchasing decisions are made at the point of sale, the challenge for a packaging proposition that stands out is critical [6]. This applies to all products and not only to the higher priced ones. On average, supermarkets have 40,000 different products, so the need to differentiate packaging into fast-moving consumer goods becomes critical [33].

## 3. QR Code as a New Buying Enabler

QR Code technology is one of the most important auto-ID technologies. It was originally designed to support automotive production lines. However, as mentioned by Kan, Teng, and Chen [34], the widespread use of mobile devices extended the use of QR Codes to many business areas, such as trade, retail, marketing, logistics, education, tourism, entertainment, etc. [35–39]. Furthermore, Tiwari argued that as consumers feel increasingly more attached to their smartphones and carry them at all times, including when they go shopping, marketers will generate new ideas and methods to better reach consumers [40].

The number of applications is much higher for QR Code tags, due to the large capacity of information they store, the high speed of reading them, and mainly due to the capability to read QR Code tag information from smartphones and tablets [41]. Additionally, it should be noted that the extensive capabilities of QR Codes to retrieve web pages or files contained in web pages offer many opportunities to support marketing activities.

Pozin considered the QR Code to be a powerful, low-cost tool that can enhance the relationship of a brand with consumers [42]. For example, QR Codes can be printed on advertising media such as magazines, newspapers, posters, or the packaging of a product. Consumers are linked through a mobile phone to a specific website and receive additional information about the product or a general advertising message for which there can also be interactivity [43]. Narang, Jain, and Roy also noted that the addition of QR Codes on print advertising offers interactivity and enables consumer tracking, such as browsing time on the site, QR Code scan frequency, and consumers' geographical location [44]. As Shamim, Xiaoyan, and Farjana argued, QR Codes help to obtain valuable information regarding consumer behavior, demographic information, and response rates [45].

The major advantage of interactive mobile marketing is its ability to provide information in a personalized and interactive way without time and place restrictions [46]. QR Codes included in product packaging, on labels, and in commercial spaces (shelves, showcases, posters, etc.), are considered particularly effective in providing timely product

and brand information given their capacity to reach consumers when and where they are ready to purchase with relevant, targeted, and interactive information [13,47,48].

The use of technology enhances packaging effectiveness as a marketing tool and its potential to increase customer engagement, satisfaction, and retention by making consumers' engagement with the product more interactive [49,50]. The level of customer engagement and trust is considered higher when companies share more information about the product and are completely transparent towards consumers (Huang, 2018). Moreover, a new trend regarding QR Codes is the customization of the QR Codes in terms of color and image use, making them a powerful branding tool that may increase brand awareness [50].

During the early years of QR Code usage in marketing activities, it was considered to be technology at an "adolescent" maturity level [39,51] with low awareness among the majority of consumers [52,53]. Today, the QR Code is a de-facto part of any packaging and especially in the food products [2]. With the use of QR Codes, Armstrong et al. expanded the definition of packaging as a container *"that contains technology (e.g., sensors, codes, and tags) capable of generating data that can be captured, treated, analyzed and communicated to people or machines to change behavior in the physical world"* [49].

## 4. Research Methodology

In the present study, the authors aimed to examine if the use of QR Code and enriched text information technology positively affect the assessment of knowledge about the product and the decision to purchase it by the consumer. In addition, this study identifies the advantages of QR Codes compared to conventional labels. The objective was to evaluate the use of web applications using enriched text information.

The current study expands the technology acceptance model (TAM), a well-known technology acceptance and adoption model, which validates the user acceptance of specific technology based on the following two variables: perceived usefulness and perceived ease of use [54]. The TAM model proposed these variables to be the fundamental determinants of user acceptance. Taherdoost (2017) as well as Alomary and Woollard (2015) made a thorough review of technology acceptance and adoption models and theories [55,56], while Röcker (2010) [57] argued that existing technology acceptance models are only of limited use for predicting and explaining the adoption of future information and communication technologies. Hence, a number of extended versions of the TAM have been extensively used for various research initiatives aiming to research consumers' acceptance of a new technology based on more variables. Indicative research studies are Alharbi (2022) for virtual reality, Islam (2011) for Internet services, Al-Mushasha (2013) for e-learning platforms, Wang and Goh (2017) for video games, Rafique et al. (2020) for mobile applications, etc. [58–63]. There are also research papers regarding the introduction of an extended TAM for measuring user acceptance of marketing communications through QR Codes [37,38,64,65].

In this paper, the authors propose an extended TAM model by adding three more variables: perceived knowledge of the product, perceived communication or interactions with the company's website, and perceived feeling of security. This constitutes the base for the conceptual framework that explores consumers' intention to use QR Codes on packaging (Figure 2).

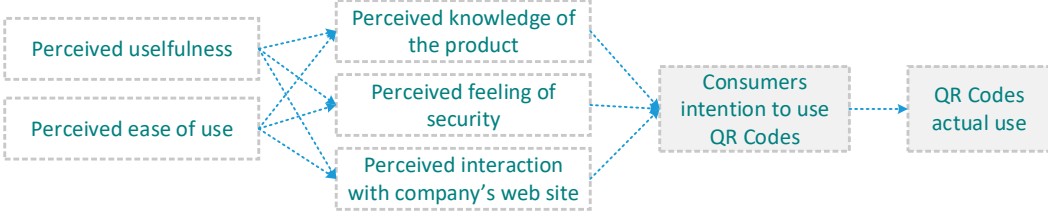

**Figure 2.** Conceptual framework.

Based on the above, the research question of this study is:

*Can QR Code and enriched text information technology positively affect the consumer's perceived knowledge of the product, perceived communication or interactions with the company's website, and perceived feeling of security?*

Specifically, we examine the effectiveness of the use of QR Codes on consumers' behavior for a specific product, the bottled milk produced by the American Farm School (AFS) of Thessaloniki, Greece, and on a specific market segment, Northern Greece. Hence, the target population of the study was consumers of milk produced by the AFS and, in order to examine the impact of QR Codes on their knowledge of the product and their purchasing decisions, two questionnaires were developed and used in the two stages of the research. The top management of AFS provided access to the customers' historical sales data as well as contact information. This is part of AFS's sales and marketing policy because the examined product has had loyal customers for more than 20 years.

We used two questionnaires as data-gathering tools. The first questionnaire referred to the milk product with the existing simple packaging and label. The second questionnaire referred to the same label with the addition of a QR Code, which linked consumers to a specific website developed for the research (https://sxolistogala.gr, accessed on 1 December 2021). The two labels (with and without a QR Code) are depicted in Figure 3 and the homepage of the linked website is provided in Figure 4.

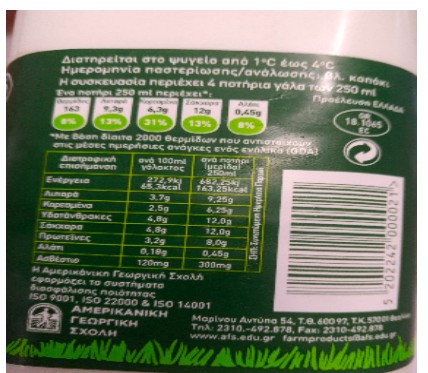 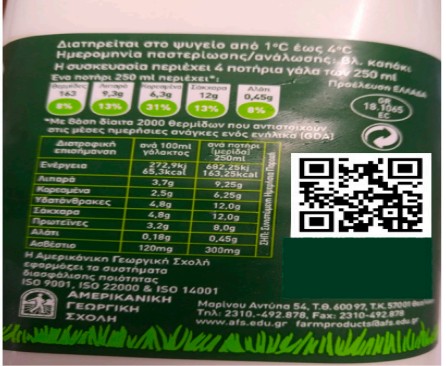

**Figure 3.** Labels without (**left**) and with (**right**) QR Code.

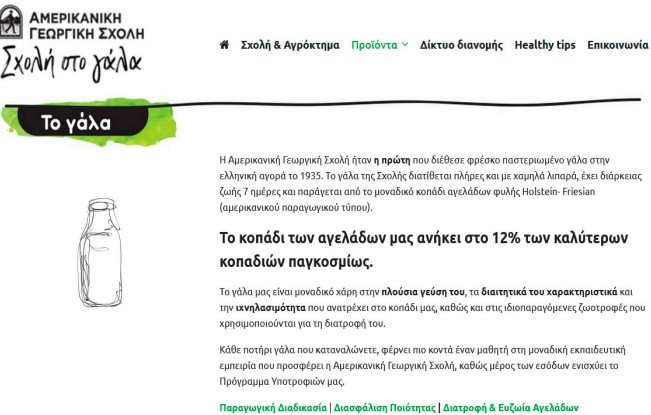

**Figure 4.** The linked website (https://sxolistogala.gr, accessed on 1 December 2021).

First, 537 customers completed the first questionnaire and after 1 month, the second questionnaire was sent to them, which was completed by 308 customers (~57% of the initial sample). The following table (Table 1) presents the two samples' demographics.

Similar frequencies and percentages are observed in terms of gender and educational level between the above samples. Younger respondents (16–35) participated in the 2nd questionnaire, more specifically, 34% versus 20.5%. In the first stage, female respondents were more likely to be university graduates ($x2$ (1) = 14.016, $p < 0.01$) and to be in the age group

26–45 ($x2$ (1) = 36.591, *p* < 0.001). Moreover, respondents in the age group 36–55 have postgraduate or doctoral degrees. Similar findings were reported in the second questionnaire ($x2$ (1) = 12.005, *p* < 0.02; $x2$ (1) = 6.628, *p* < 0.04; $x2$ (1) = 22.950, *p* < 0.01) correspondingly.

**Table 1.** Respondents' demographics.

|  |  | Without QR Code (N = 537) | With QR Code (N = 308) |
|---|---|---|---|
| Gender | Men | 237 (44.1%) | 133 (43.2%) |
|  | Women | 300 (55.9%) | 177 (56.8%) |
| Age group | 16–25 | 25 (5.0%) | 26 (8.4%) |
|  | 26–35 | 83 (15.5%) | 78 (25.3%) |
|  | 36–45 | 224 (41.7%) | 90 (29.2%) |
|  | 46–55 | 138 (25.7%) | 84 (27.3%) |
|  | >55 | 65 (12.1%) | 30 (9.7%) |
| Education level | Secondary | 46 (8.6%) | 28 (9.1%) |
|  | Tertiary | 226 (42.1%) | 119 (38.6%) |
|  | Postgraduate or doctoral degrees | 265 (49.3%) | 161 (52.3%) |

## 5. Findings

As mentioned above, the targeted product has had a loyal customer base for the last 20 years. Indeed, according to the responses, approximately 2 out of 3 consumers (305, 67%) buy the American Farm School milk once or more times per week. A total of 398 (~75%) stated that they consciously use the food packaging to make a purchase decision, while approximately 1 out of 3 (199, 37%) are either very or very much affected by the packaging in their decision to purchase the product.

In the next part of both questionnaires, there were questions that aimed to measure consumers' perceived knowledge of the product based on the content that is provided by the label (1st questionnaire, without QR Code) and by the website (2nd questionnaire, with QR Code). Specifically, they included questions such as:

- Do I need to know anything before I open the package?
- Is the product Greek?
- Is the product organic?
- Are the pasteurization/consumption dates listed on the bottom of the package?
- Is the label made from recyclable materials?
- Does the American Farm School implement quality systems?
- Does the package contain 4 glasses of milk?
- Does the label contain contact information?
- Does the label contain nutritional tips?
- Does the label contain the number of calories?

The 2nd questionnaire included some additional statements, such as:

- The American Farm School provides training seminars on the farm for milk production.
- The site presents recipes.
- Milk has a shelf life of 7 days.
- Milk is available daily on the market less than 24 h after milking.
- The website contains contact information.
- The website contains nutrition tips.

Consumers had three response options: "Agree", "Disagree", and "Don't know". For all questions, there was only one correct answer. According to the responses, QR Code technology helped consumers better identify and evaluate the characteristics of a product compared to the conventional label. This was due to the number of correct answers as well as the number of "Don't know" responses for both questionnaires. *t*-test results for the two samples reveal that the mean for the correct answers "With QR Code" was higher than those "Without QR

Code" (9.32 > 8.66), the "Don't know" answers were lower "Without QR Code" (1.86 > 2.37), and both differences were statistically significant (df = 843, $p < 0.001$).

The method of analysis of variance (ANOVA) was applied in order to evaluate the effect of the selected factors on the number of correct answers (Table 2). The factors included in the model were Gender, Age Group, Education Level, Frequency, and With or Without QR Code. The factors Age Group ($p = 0.637$) and Education Level ($p = 0.999$) were found to be statistically non-significant. On the other hand, the factors Gender ($F_{(1, 295)} = 4.355$, $p = 0.038$), Frequency ($F_{(4, 295)} = 3.876$, $p = 0.004$), and With or Without QR Code ($F_{(1, 295)} = 5.617$, $p = 0.018$) were found to have a statistically significant effect on the number of correct answers.

**Table 2.** Factors affecting the number of correct answers (ANOVA).

| Variable | SS | F | df1 | *p* |
|---|---|---|---|---|
| Intercept | 4275.625 | 993.094 | 1 | 0.000 |
| Gender | 18.748 | 4.355 | 1 | 0.038 |
| Age Group | 10.958 | 0.636 | 4 | 0.637 |
| Education Level | 0.011 | 0.001 | 2 | 0.999 |
| Frequency | 66.756 | 3.876 | 4 | 0.004 |
| With or Without QR Code | 24.185 | 5.617 | 1 | 0.018 |
| Residual | 1270.081 | | 295 | |

In the context of the tested model, men were found to give significantly less correct answers than women did (6.69 ± 2.24 vs. 7.16 ± 2.06, $p = 0.038$, Figure 5).

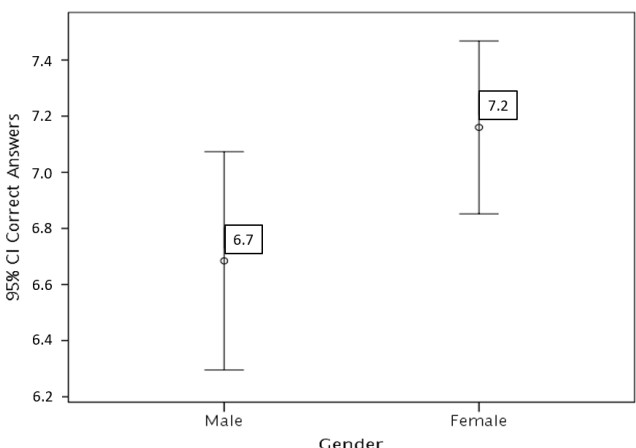

**Figure 5.** Gender effect on the number of correct answers.

Concerning the Frequency influence on the number of correct answers, the following significant differences were found (Figure 6):

- "Seldom" (6.44 ± 2.32) < "Frequently" (8 ± 1.52), (Diff = 1.564, 95% CI 0.442–2.687, $p = 0.001$).
- "Seldom" (6.44 ± 2.32) < "Sometimes" (7.50 ± 1.98), (Diff = 1.064, 95% CI 0.179–1.950, $p = 0.010$).
- "Rare" (6.78 ± 2.08) < "Frequently" (8 ± 1.52), (Diff = 1.221, 95% CI 0.055–2.387, $p = 0.035$).

Concerning the presence of QR Code influence on number of correct answers, the following significant differences were found:

- "Without QR Code" (6.40 ± 2.40) < "With QR Code" (7.28 ± 1.94), (Diff = 0.874, 95% CI 0.390–1.357, $p = 0.001$).

Furthermore, participants who used a QR Code had significantly more correct answers than the participants who did not use a QR Code (Figure 7, 7.28 ± 1.94 vs. 6.39 ± 2.42, $p = 0.001$).

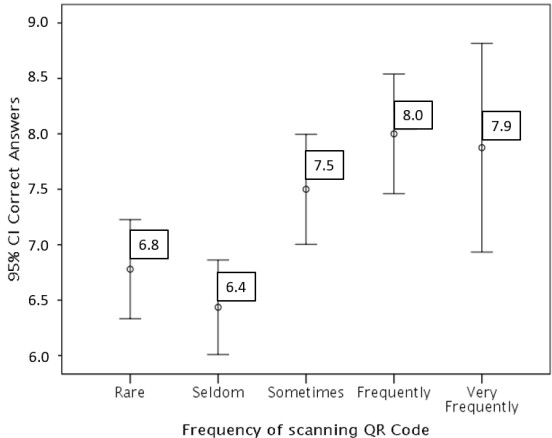

**Figure 6.** Frequency of scanning effect on the number of correct answers.

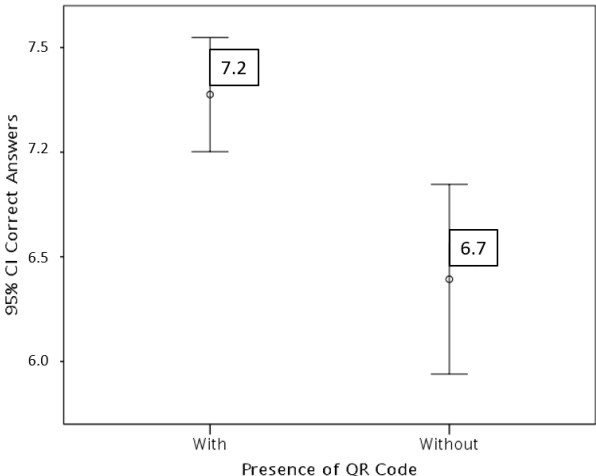

**Figure 7.** Presence of QR Code effect on the number of correct answers.

*Factors Affecting the Number of "Don't Know" Answers*

Following this, the method of analysis of variance (ANOVA) was applied in order to evaluate the effect of the selected variables on the number of "Don't Know" answers (Table 3). The factors used in the model were Gender, Age Group, Education Level, Frequency, and With or Without QR Code. The factors Age Group ($p = 0.811$) and Education Level ($p = 0.938$) were found to be statistically non-significant. On the other hand, the factors Gender ($F_{(1, 295)} = 4.945$, $p = 0.027$), Frequency ($F_{(4, 295)} = 3.231$, $p = 0.013$), and With or Without QR Code ($F_{(1, 295)} = 12.274$, $p < 0.001$) were found to have a statistically significant effect on the dependent variable.

**Table 3.** Factors affecting the number of "Don't Know" answers (ANOVA).

| Variable | SS | F | df1 | *p* |
|---|---|---|---|---|
| Intercept | 435.576 | 109.312 | 1 | 0.000 |
| Gender | 19.706 | 4.945 | 1 | 0.027 |
| Age Group | 6.323 | 0.397 | 4 | 0.811 |
| Education Level | 0.507 | 0.064 | 2 | 0.938 |
| Frequency | 51.491 | 3.231 | 4 | 0.013 |
| With or Without QR Code | 48.906 | 12.274 | 1 | 0.001 |
| Residual | 1175.484 | | 295 | |

Concerning the Gender influence on the number of "Don't Know" answers, in the context of the model, men were found to give significantly more "Don't Know" answers than women (2.15 ± 1.98 vs. 2.64 ± 2.21, $p$ = 0.027, Figure 8).

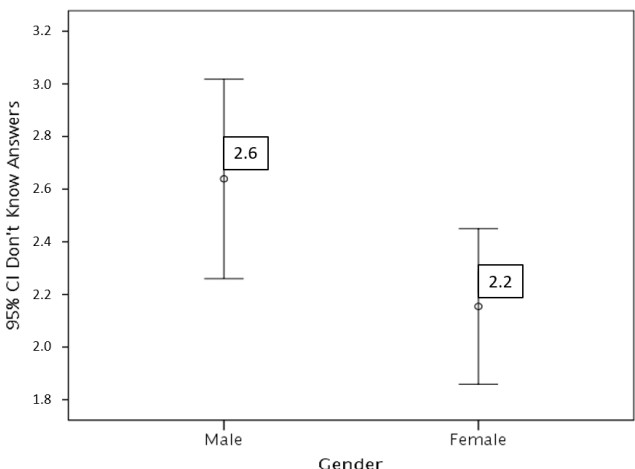

**Figure 8.** Gender effect on the number of "Don't Know" answers.

Concerning the Frequency influence on "Don't Know" answers, the following significant differences were found (Figure 9):

- "Sometimes" (1.61 ± 1.63) < "Seldom" (2.83 ± 2.32), (Diff = 1.220, 95% CI 0.368–2.072, $p$ = 0.001).
- "Sometimes" (1.61 ± 1.63) < "Rare" (2.63 ± 2.13), (Diff = 1.019, 95% CI 0.114–1.923, $p$ = 0.018).
- "Frequently" (1.70 ± 1.55) < "Seldom" (2.83 ± 2.32), (Diff = 1.132, 95% CI 0.052–2.212, $p$ = 0.035).

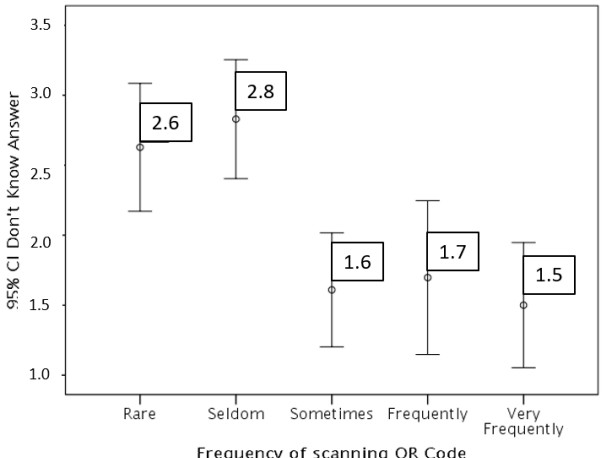

**Figure 9.** Frequency of scanning effect on the number of "Don't Know" answers.

Furthermore, participants who used a QR Code had significantly fewer "Don't Know" answers than the participants who did not use a QR Code 1.95 ± 1.72 vs. 3.08 ± 2.47, $p$ = 0.001) as depicted in Figure 10.

Finally, in the last part of the 2nd questionnaire (With QR Code) consumers were asked to evaluate how the perceived communication or interactions with the company's website and the resulting feeling of security affected their perception of the product usefulness and ease of use.

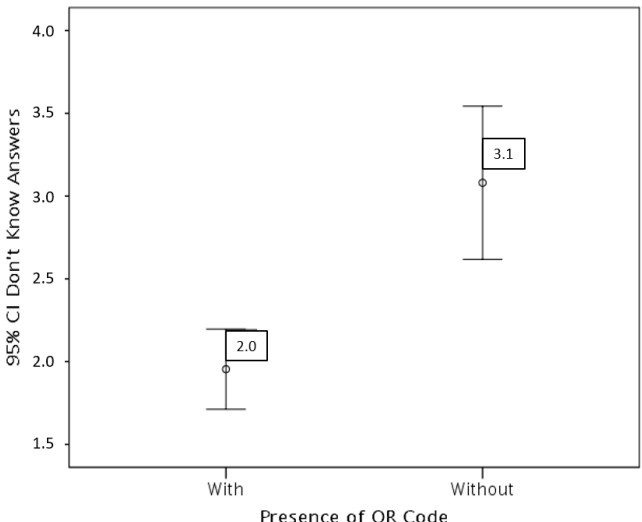

**Figure 10.** Presence of QR Code effect on number of "Don't Know" answers.

The results show that the use of the QR Code on the label of the milk bottles and the visit to the site (1) made the process of obtaining information on the product easier for 87.2% of the consumers, (2) helped them receive better information about the product for 83.7%, (3) made the process of obtaining information about the product more efficient for 85.7%, and (4) helped respondents get better information on the American Farm School's activities for 78.1%.

Overall, consumers who scanned the QR Code and visited the website gave more correct answers and less "Don't Know" answers to the questions about the product.

Of great importance are the responses to the following questions, answered by all respondents:

- Do you believe that: The use of the QR Code is an easy process?
- Do you believe that: The interaction regarding the QR Code is clear and comprehensible?
- Do you believe that: The QR Code will affect my future buying decisions? and
- Do you believe that: I will be using the QR Code more often in the future?

A total of 4 out of 5 respondents consider the QR Code to be an easy-to-use procedure with the respective interaction to be a clear and comprehensible task, while 1 out of 2 will use the QR Code whenever they want to obtain more information on a product. However, only 2 in 5 believe that the QR Code will (positively or negatively) influence their future purchasing decisions.

In regards to the arguments highlighted by previous research [39,50–52] on the reasons that discourage users from using the QR Code, the sample's responses suggest that the use of the QR Code is safe (74.7%) and it cannot cause leakage of sensitive personal data (61.4%). It cannot mislead consumers (74.4%) and it constitutes a useful technology that effectively facilitates food product purchases (81.5%), further reinforcing the research presented in the 3rd section of the paper.

## 6. Conclusions

Today, more than ever, businesses are realizing that packaging can strongly influence consumer decision-making and also improve business performance in storage and transportation by standardizing their respective logistics activities.

In this study, we aimed to investigate the degree of influence of QR Codes on consumers' shopping habits by collecting data from a structured questionnaire and for a specific product, the bottled milk produced by the American Farm School of Thessaloniki. In this context, primary research was conducted to clarify the benefits arising from the use of QR Codes to understand the relationship that develops with consumers who are familiar with new technologies. Since most respondents of this study in the segment "With

a QR Code" had postgraduate or doctoral degrees, we can argue that the sample is not representative but a number of interesting findings and managerial implications emerged from the study.

Two-dimensional (2D) QR barcodes were originally intended to replace or supplement the classic barcodes, which, due to their limited size, cannot store large volumes of information and are therefore unable to meet the modern needs of digital tools and the Internet. Essentially, QR Codes are a way to map the available information into a graph, which can then be read and scanned, both by standard readers, such as barcode scanners, and by smartphones and devices (such as tablets) that now have high-definition cameras suitable for reading such graphics. The use of smartphones makes this technology particularly attractive to businesses and consumers, allowing the targeted dissemination of "information" as a tool of mobile marketing [35,66].

The findings of the primary research demonstrated the importance of packaging in food both with and without QR Codes. Properly designed packaging can reduce unnecessary costs and resources. This may reduce the selling price of the product, making it more competitive. It can also lead to increased levels of efficiency and effectiveness of the whole supply chain. Finally, packaging provides the appropriate information to the consumers. Thus, it functions as a means of communication and product promotion. The consequence of this is increased sales.

The usage of the QR Code, in particular, helps consumers gain better knowledge regarding the product, thus providing a clear answer to the second research question. Rich and interactive information helps consumers better comprehend the product characteristics, even in the case of a product to which they are loyal (such as the one investigated in this research) [67].

The results showed that, by scanning the QR Code and going to a dedicated website, the consumer gives correct answers about the appreciation of the product while, at the same time, the answers of "Don't know" are significantly reduced (in other words, consumers show confidence in their answers). Research has shown that comprehension and self-confidence are higher with the adoption of the QR Code in accordance with other studies [66,67]. The QR Code also helps the business itself to provide accurate information and positively influence consumers' buying behavior [68].

The use of QR Codes on packaging presents specific advantages over the traditional packaging, thus enhancing the purchasing process, the information obtained regarding the product, and the organization that produces and markets it. This technology provides a greater volume of information to consumers compared to what can be written on a label. Regarding the ease of use of QR Codes, participants in this research did not appear to have any problems while, at the same time, they felt that the use of QR Codes made it easier for them to read the product-related information because of the font size adjustment function and the provision of rich information [11,67].

Based on the above findings, we suggest that the producer of bottled milk, in this case, the American Farm School, should consider the introduction of the QR Codes technology. In general, by using the dynamic QR Code, producers can monitor the scanning activity (track marketing, logistics, and traceability data), support analytics, and optimize dual-platform advertising. This was highlighted in various studies in the targeted sector [17,69,70]. Moreover, consumers can get all the information they want regarding a product via their mobile phones and the corresponding website that will derive more functionalities in the near future. QR Codes serve consumer values through traceability. Any food information can be tracked with a QR Code, as previous studies appreciate [71,72]. As Müller and Schmid (2019) pointed out *"These systems can monitor permanently the quality status of a product and share the information with the customer"* [72]. That includes a product's source history: a running account of where it originated, was manufactured, and was distributed. That, along with relevant information about the production, manufacturing and distribution facilities will digitize the food production procedure, supporting public policy initiatives to make information accessible, traceable, and verifiable by the consumers and producers [73].

In conclusion, QR Codes are here to stay. Companies in the food sector can design and offer a personalized experience for customers who visit their operations, whether physically or electronically while, at the same time, they can influence their buying behavior [74,75]. Future research should evaluate the efficiency of QR Codes on food packaging based on the expectations of industry managers.

**Author Contributions:** Conceptualization, K.R. and D.F.; methodology, A.K.; validation, C.B. and L.H.; investigation, D.F.; resources, K.R.; writing-original draft preparation, A.K.; writing-review and editing, T.F., L.H. and C.B.; visualization, A.K.; supervision, T.F.; project administration, T.F. All authors have read and agreed to the published version of the manuscript.

**Funding:** This research received no external funding.

**Data Availability Statement:** Not applicable.

**Conflicts of Interest:** The authors declare no conflict of interest.

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
