# Peer review of "Evaluating the Use of QR Codes on Food Products"

_sustainability, doi:10.3390/su14084437_

Round 1

Reviewer 1 Report

The paper was slightly improved. It seems that explanations about each reviewer’s comments are missing. Methodology is better detailed in this new version.

Results have managerial implications, and authors could have explored opportunities for public policy as well. Theoretical contributions are still not that clear, although it seems that the study has a more managerial oriented focus. Given that, I believe this study is publishable if the editor understands that the scope of the journal covers research not theoretically-driven.

Author Response

Attached are our responses to your useful comments/suggestions.

Reviewer 2 Report

The paper brings interesting research results, but in the context of a scientific work and analysis it is necessary to refine certain parts of the paper.
The results and discussion should be compared with similar research.
The structure of the research is not well defined methodologically. The research questions need to be asked and explained in the research methodology. It is not known how the respondents were selected and it is not stated. It is also necessary to explain more clearly the parts of the comparison of answers.
It is not clear how the results are presented, e.g.
"Approximately 2 out of 3 consumers (305, 67%) who responded in the first questionnaire purchase American Farm School milk once or more per week. "

Author Response

(The authors gave the same response as above.)

Reviewer 3 Report

The manuscript is clear, well-written and relevant for the field.

Author Response

(The authors gave the same response as above.)

Reviewer 4 Report

Finding parts need to be support by previous research. This shows more quality of the research.

Author Response

Attached are our responses to your useful comments/suggestions.

This manuscript is a resubmission of an earlier submission. The following is a list of the peer review reports and author responses from that submission.

Round 1

Reviewer 1 Report

Authors identified an interesting topic, but it seems that the research lacks a more robust methodology. The study aims to investigate the influence of QR Codes in food packages on the enhancement of the information provided to consumers about the product and purchasing intention. An online experiment could have provided more valuable insights to investigate the impact of a QR Code on purchase intention, consumers` perceptions about the product and other dependent variables. It seems that a conceptual framework is missing in this study to better illustrate which variables were investigated in relation to the potential impacts generated by the introduction of a QR Code. Authors mention that the degree to which QR Codes on bottle milk packaging increases the purchase intention was accessed, but it is not clear in the results how the purchase intention was measured and to which extent the QR Code impacted on it.

It would be interesting if authors could have access to data from the food industry to check how many consumers access information about food products via QR Codes in the package.

The introduction needs to be rewritten to present the research gaps and the theoretical contribution of this study. From the introduction, the reader should clearly see the research gaps that the study seeks to bridge. Other possibility would be to present potential managerial implications of this study.

 Most respondents of this study in the segment “With a QR Code” had postgraduate or doctoral degrees, as such authors could highlight that the sample is not representative.

Conclusions need to abstract more from the data gathered. Authors could reflect more on the data and make links with managerial implications and the theoretical contributions.

Author Response

We would like to thank you for the time and effort you devoted to studying our paper. Your extensive comments and suggestions have greatly helped us to significantly improve our work. In addition, some of your suggestions, although not possible to be implemented in this paper, gave us new ideas for further research on this topic in the future and we greatly appreciate that.

Reviewer 2 Report

The idea of the paper is really interesting and presented results also. For the better understanding of all research and paper I would suggest that you maybe consider to do some improvements:

In chapter materials and methods just put the questions which you used in this research - you are mention those questions on the end in the chapter 5.3. It will be much easier to follow manuscript. 

In the second chapter you are presenting the five main elements of the relationship between packaging and marketing. This all text is more stable for the introduction. 

Conclusion should be answer to your objective and not so long. In conclusion we do not usually cited other authors. It is more for the Discussion.

In general, the paper is really interesting and I my opinion its that with this reorganisations in the tex you will get more readable text. 

Author Response

(The authors gave the same response as above.)

Reviewer 3 Report

English language and style are fine/minor spell check required

Author Response

(The authors gave the same response as above.)
